# Restorative Effects of Observing Natural and Urban Scenery after Working Memory Depletion

**DOI:** 10.3390/ijerph20010188

**Published:** 2022-12-23

**Authors:** Menno van Oordt, Kim Ouwehand, Fred Paas

**Affiliations:** 1Department of Psychology, Education, and Child Studies, Erasmus University Rotterdam, 3062 PA Rotterdam, The Netherlands; 2School of Education/Early Start, University of Wollongong, Keiraville, NSW 2522, Australia

**Keywords:** attention restoration theory (ART), working memory capacity (WMC), depletion, natural environments, urban environments, cognitive restoration, picture viewing

## Abstract

According to attention restoration theory observing nature has restorative effects on cognitive components, such as working memory, after a cognitive depleting task. Additionally, urban environments are thought to have no effect or even a negative effect on cognitive restoration. Previous research has confirmed that observing actual, as well as digitally presented nature sceneries leads to more restoration of working memory capacity (WMC) than observing (digital) urban sceneries. To further investigate these findings, we conducted an experiment with 72 university students as participants. After a WMC depleting task, participants observed either digitally presented nature scenery, urban scenery or no scenery, and subsequently performed a digit span test, which was used to measure restoration of WMC. Results indicated significant higher performance on the digit span test for those who observed nature scenery in comparison to those who observed urban scenery or no scenery, thereby replicating results from previous research. Observing urban scenery was neither harmful nor helpful in terms of cognitive restoration compared to observing no scenery. These findings provide a foundation for implementing a brief intervention of observing nature in academic settings to facilitate the restoration of WMC.

## 1. Introduction

Over the last decades, an increasing body of evidence has emerged on the benefits of interacting with nature. Positive effects of interacting with nature have been found on physical health, psychophysiological processes, mental health and cognition (e.g., [1,2,3]). Further evidence is provided in a recent review by Lackey et al. [4] that showed reductions in anxiety, stress and depressive symptoms, and improvements in emotions, mental well-being and cognition after interacting with nature. Concerning the latter, research on the positive effect of nature on cognition can be found in multiple studies. A study by Schutte et al. [5] for example, showed how children responded faster on attention tasks after walking in nature, compared to walking in an urban area. Another study showed that compared to a walk in an urban area, a walk in nature had positive effects on verbal working memory [6].

In the present study it was investigated whether observing nature, compared to urban scenery, would enhance working memory restoration after being depleted of cognitive resources. We will start by reviewing the theoretical and empirical evidence supporting the effects and underlying mechanisms of interactions with natural and urban environments on cognition, specifically working memory.

### 1.1. Attention Restoration Theory in Natural and Urban Environments

Studies investigating cognitive benefits of interacting with nature commonly use the attention restoration theory (ART) [7,8]. The ART proposes a restorative mechanism of cognition. This mechanism relies on two components of attention; involuntary attention and directed attention. Involuntary attention, according to the work of James [9] is attention, involuntarily captured by environmental stimuli. Directed attention refers to attention which is controlled by effortful (voluntary) cognitive processes. Studies have shown how cognitive processes are prone to depletion. For example, research has shown that especially cognitive effortful tasks can deplete cognitive resources, and can cause underperformance on subsequent cognitive tasks following the effortful tasks [10,11,12]. The ART uses these mechanisms described by Kaplan [7] and James [9] to explain the restorative effects of nature on cognition, especially directed attention. Observed stimuli from natural environments usually evoke effortless involuntary attention, thereby allowing directed attention mechanisms a chance to restore [7]. This has led to the idea that after restoration of directed-attention mechanisms, one could perform better on directed attention or effortful tasks after interacting with nature. In contrast, urban environments can impose a heavy demand on directed attention processes according to the ART. Urban environments, such as traffic or the ambiance of a shopping center provide more stimuli which capture attention. Directed attention is then required to remain oriented, by for example avoiding traffic or following a certain route. This sets forth the idea that urban environments may in contrast be demanding on directed attention and therefore have no significant restorative effects on cognition, or might even hinder the restorative effect (e.g., [13,14]).

Quite some research has focused on the restorative effects of nature on cognition, typically by having participants walk in or observe nature. A common finding is that effortful cognitive processes and emotional affect benefit from being in nature, in contrast to being in urban environments. One prominent example is a study by Berman et al. [1]. In one of their experiments, participants made a digit span task as cognitive measure of working memory capacity. Afterwards, a directed-forgetting task in which participants were requested to forget (suppress) specific information was made which was meant to cognitively fatigue the participants. Then, participants either took a 50-to-55-min walk in an urban city center, or in a nearby park. Afterwards, each participant would make the digit span task again to conclude whether walking in the park would be beneficial for cognition, contrary to walking in the city center. Results revealed significant improvements in performance on the digit span task for participants who had walked in the park, and not for participants who had walked in the city center, even when controlling for mood and weather conditions. Similar positive cognitive effects of walking in- or observing nature have been found in other studies (e.g., [5,6,15]).

Complementing the approach of physically presenting nature to benefit cognitive restoration, other studies on the same topic have used computer-based environments to provoke the same effects. In general, similar restorative effects on cognition have been found in these studies, in which participants viewed the nature and urban environments through pictures or videos. In a study by Gamble et al. [13] for example, both older and younger adults took a digit span task and part of the attention network task as cognitive measures. The attention network task measures three types of attention processes; alerting, orienting and executive attention. Whereas alerting and orienting are more stimulus-driven, executive attention is a more effortful and top-down function in which selective and inhibitory processes are involved. After completing the tests, participants either watched urban pictures or nature pictures for five minutes depending on the condition in which they had been placed. Following the picture viewing, participants made another part of the attention network task and took the digit span task for a second time. Results indicated that there was a significant improvement in executive attention for both older and younger adults after viewing natural pictures, but not urban pictures. No significant difference was observed in the digit span assessment when comparing urban and nature pictures, other studies have. Berman et al. [1] compared participants who viewed natural pictures to participants who viewed urban pictures, and found a significant improvement in digit span score for the former group. Among those described previously, other studies investigating the positive effects of observing computer-based natural environments on cognition have found similar positive results (e.g., [16,17,18]). Whereas, some research suggests that in some cases, actual nature may be more beneficial for restoration compared to digitally presented nature, both types of exposure to natural environments have been found to produce positive effects on cognition [19]. Although the ART suggests that observing urban scenes may be demanding on cognition, no studies have found a cognitive fatiguing effect of observing urban environments.

A study by Mason et al. [20] ties single studies on the restorative effects of nature together in a literature review. Their review focused on the short-term effects of interacting with nature on cognitive restoration and cognitive performance across different educational levels. Out of the fourteen studies included in their review, only two did not induce cognitive depletion before having the participants come in contact with nature. Interestingly, these were the only two studies which did not reveal cognitive benefits. This might indicate that the interaction with nature to restore cognitive functioning is only beneficial after effortful cognitive processes. Overall, their review concluded positive results regarding cognitive restoration due to brief interactions with nature (10 min up to 90 min), after being depleted of cognitive resources. These benefits were found in elementary school, high school and higher education [20].

### 1.2. Alternative Explanations

Beside the framework of the ART, there are other theories which attempt to explain the restorative effects of nature. Ulrich [21] proposed the stress reduction theory (SRT) with an innate evolutionary perspective, as opposed to the ART which is based on a cognitive perspective. The SRT proposes that humans have evolved and developed in natural environments as opposed to urban environments. The SRT is based on the biophilia hypothesis, proposed by Wilson [22]. The hypothesis states that since humans have evolved and lived in nature, we have an innate need to connect with it. Humans therefore tend to be more comfortable in nature, where we lived for thousands of years. With this in mind, Ulrich argues that being in natural environments, or viewing natural scenes would activate an evolutionary emotional response; a decrease in heart rate and blood pressure which in turn evoke sustained attention and offer a chance for cognitive restoration. In contrast to natural environments, the SRT also explains that stimulating urban environments can affect people negatively through the presence of stressful and constraining levels of physiological and psychological arousal [21,23].

The ART and the SRT explain the restorative effects of natural environments, and the possible restraining effects of urban environments from different perspectives. Both theories however, share the common idea that nature can in turn restore cognitive resources. Both theories, first suggested in the 1980s, have received quite some support over the years along with critique, both still have considerable value today [24]. 

### 1.3. ART and Executive Functions

Effortful cognitive processes, which especially according to the ART could be restored by observing natural stimuli, can also be referred to as executive functions. Executive functions (EF) are effortful cognitive functions such as executive attention or working memory [25]. Executive functions stand in close relation to the attention restoration theory framework proposed by Kaplan [7]. Kaplan and Berman [26] extended this framework by describing the relationship between directed attention and executive functions. The authors state that directed attention, which requires effortful top-down processing, can be seen as a common resource for executive functioning, and that executive functions are therefore also prone to depletion. It is hence not uncommon that measurements of executive functions are used in attention restoration research as effect measure (e.g., [1,5]). The working memory, as an executive function will be a core aspect of the current research. The working memory, as described by Diamond [25] is involved in mentally holding information, and working with or manipulating the information when it is no longer perceived. Working memory is critical in many mental activities, such as mental calculation, mentally ordering and recalling items and any forms of incorporating information into ongoing thought processes.

### 1.4. ART, EF and Academic Performance

Research evidence has shown that executive functions play an important role in learning and academic performance. Even when controlling for intelligence, different executive functions could positively predict academic performance [27]. Focusing on the executive function of working memory, Chalmers and Freeman [28] showed that performance on a working memory task was positively associated with the NAPLAN, an Australian national curriculum assessment in children just under 10 years old. Another study by Rhodes et al. [29] revealed a positive correlation between working memory and performance in chemistry in early high school children. Given these findings, it may be beneficial to examine whether executive functions, especially working memory can be restored using natural environments. Even to the extent of improving academic performance. 

The working memory is thought to be prone to depletion after effortful tasks, such as learning and regular classroom activities. Investigated in a study by Chen et al. [11], students engaging in massed practice of algebraic materials suffered more from working memory depletion than those who spaced their learning, and this could explain the differences in their learning outcomes. Though instructional design or the spacing effect is not the goal of the current study, the finding from Chen et al. [11] does show how working memory can be depleted in a classroom setting. For this reason, working memory will be the main focus of this study.

The trend to go into nature and restore cognition may not be easily accessible to everyone. Therefore, the opportunity to research artificial environments (e.g., viewing pictures on a screen) to improve cognitive restoration could be a desirable alternative. Given the relationship between executive functions and academic performance [27,28], investigating the attention restoration theory framework in classroom settings is a relevant research topic. Although it would be difficult to teach classes in a physical nature setting, filling a classroom with natural stimuli could be a possible way to achieve the restorative effects of nature. Research on the ART in the classroom has been investigated before by Moreno et al. [30], who developed an app based on the principles of the ART. The app would play a randomly selected nature video lasting two minutes. Frequency of use of the app was positively related with academic scores. The connection between app usage and executive functions however, could not be made. 

### 1.5. Research Question and Hypotheses

Previous research on the effects of observing natural vs. urban scenery mainly focused on restoration of depleted attentional resources in the context of the ART [8]. The present study will focus explicitly on restoration of depleted working memory resources. Other studies that did look at working memory resource depletion in the context of other theoretical frameworks, did not use natural and urban scenery to influence restoration [11,12]. Additionally, in previous research on the attention restoration effect researchers have compared effects of natural and urban scenery, but not used a non-scenery control group. Consequently, it is not possible to tell whether the observed effects were due to the restorative effects of natural scenery, the possible hindering effects of urban scenery, or both e.g., [1,13]. Therefore, the present study can be argued to advance the field by investigating restoration of working memory resources, and by comparing effects of a natural-scenery, an urban-scenery, or a non-scenery control condition. 

The present study will serve as a foundation of implementing the ART framework in an academic environment. The possible positive effect of observing natural stimuli through a digital screen on the executive function of working memory will be investigated. Additionally, since the ART suggests that observing urban scenery requires involuntary attention, and hence might hinder cognitive restoration, this study will also investigate whether observing urban scenes may further fatigue the working memory capacity. 

Taken together, the research question we aim to answer is: 

What are the immediate effects of observing natural and urban environmental pictures through a computer screen on working memory capacity after a working memory depletion task?

To answer this research question we administered a cognitive depleting task to participants, and subsequently asked participants to observe sets of images of natural or urban sceneries presented on a computer screen, or no sceneries (i.e., control group). Then, working memory was assessed on the digit span test to investigate whether exposure to digital natural or urban environments could replete or hinder the repletion of the working memory capacity. It was hypothesized that after a working memory depleting exercise, exposure to natural stimuli would result in higher working memory performance compared to working memory performance after observing urban images, or observing no images (control). 

## 2. Materials and Methods

### 2.1. Participants

Taking a medium effect size of *d* = 0.50 into account (found in, e.g., Norwood et al., 2019; Schutte et al., 2016), a sample size of 66 participants was needed for a statistical power of 0.95 using the G* Power 3.1 tool [31]. Eventually, a sample of *n* = 72 university students (50 female, 22 male; Mage = 21.7 years, *SD* = 2.45) have been gathered from different majors, with a majority of 58 psychology students. Whereas the psychology students received course credit for participating in the experiment, students from other majors participated voluntarily (without receiving compensation). Participants were randomly allocated to the two experimental conditions and the control group using random allocation software [32]. 

### 2.2. Materials and Tasks

All materials and tasks were in the form of scripts that were ran through the Inquisit experimentation software [33]. Through the Inquisit software, a digital version of the Trier Mental Challenge Test [34] was presented, as well as the nature or urban pictures for the experimental groups, and a digital version of the digit Span test [35].

The Trier Mental Challenge Test (TMCT) [34] was used to induce cognitive depletion of the working memory capacity. The TMCT is a version of the Trier Social Stress Test (TSST) used to induce cognitive fatigue [36]. The TSST (and variations) is a highly standardized instrument, designed to induce consistent and reliable physiological and psychological stress resulting in cognitive depleting effects [37]. Due to the literature surrounding the TSST, equal levels of cognitive fatigue are expected across the three conditions. Using a similar setup, Li and Sullivan [38] used the TSST to induce cognitive depletion and similarly, and the digit span test as the main measure of effect. In the TMCT, participants were presented a basic arithmetic task on their screen (e.g., 5 + 7) and had five seconds to answer. Answering could be done by pressing buttons with the computer mouse or touchpad, which were labeled 0–9 on the screen surrounding the arithmetic problem. Arithmetic problems increased in difficulty over time, but could also decrease in difficulty depending on the performance of the participant. The difficulty in arithmetic problems was based on performance. A participant gained one point for answering correctly, but also lost a point for answering incorrectly or too late. Difficulty increased in four steps after gaining 36, 40, 45, and eventually 50 points. Difficulty was also decreased when a participant would go below one of these thresholds by losing points. This task took approximately 10 min to complete. No data were obtained from this task, as it was only used as stressor task to induce cognitive depletion.

The digit span test was used measure working memory capacity, which was the main measure of effect in the current study. The digit span has previously been used in research regarding the effects of urban and natural environments on cognition (e.g., [1,13,39,40]). The digit span assessment traditionally consists of two separate tests; the forward and backward trials. In the present research, both the forward and backward assessment was used, as they are thought to be involved in different mechanisms used by the working memory, thereby covering a wider range in working memory assessment [41,42]. The digit span test ran for a maximum of 14 trials on the forward condition, where each trial consisted of an increasing set of numbers, two sets per trial. In the forward condition, participants had to recall the digits presented in the digit span test in the same order as the digits were presented. Participants recalled the numbers seen on screen by selecting each number in subsequent order using the mouse. The backward condition was identical, but participants now recalled each set of numbers subsequently backwards. In both the forward and backward condition, participants were advanced to the next trial by correctly recalling at least one set of numbers presented in a trial. When a participant failed to recall both sets of numbers in one trial, the test was stopped and the data were recorded. In this set-up the digit span took roughly 15 min to complete, including forward and backward trials. When the digit span test was conducted, participants first made the digit span forward trials, and then the digit span backward trials. This is the same order as presented in the Wechsler Adult Intelligence Scale (WISC-IV) [43]. This order is also used in other research where both the digit span forward and backward are used to assess working memory capacity (e.g., [44,45]).

Two sets of materials were created for the experimental conditions:(1)Images from natural scenery, representative for a Dutch natural environment setting. Multiple images were used from a park in a Dutch city.(2)Images from urban scenery, representative for a Dutch city environment. Once again, images were used from several residential areas in a large city in the Netherlands.

Both sets consisted of 30 pictures, which were taken with a 12-megapixel digital camera, with a focal length of f/1.6 and a 26 mm wide angle lens. Pictures were cropped to an aspect ratio of 1:1 (square format) to ensure similarity of all pictures. All pictures had the same resolution of 2048 × 2048 pixels. Pictures were taken in a first-person perspective from approximately 180 cm above the ground. Additionally, both sets of images were created under the same weather conditions, as different types of weather may influence the mood of participants (e.g., [46,47]). The weather could be described as partially sunny, with no precipitation. Samples of the pictures that were used in both the natural and the urban setting can be observed in Figure 1. The entire set of nature pictures can be observed in Appendix A, the entire set of urban pictures can be observed in Appendix B.

### 2.3. Procedure

Participants were approached through an e-mail, which contained basic information and instructions regarding the experiment. Within the e-mail, participants could find a link through which they could participate in the experiment. Therefore, all participants were able to engage with the experiment from home. When participants entered the study, they were asked to read and sign the informed consent form. The consent form informed participants about the experiment, stating the basic contents of each numerical task as well as the length and the order in which they would be presented. In the natural and urban context groups, participants were also informed that they would be asked to merely view pictures after the first numerical task. Additionally, participants were informed that participating in the study would involve no more risks to physical or psychological health beyond those encountered in the normal course of everyday life. At last, the consent form stated that the collected data would be kept confidential and that participation was voluntary; participants could withdraw from the study at any time without further consequences. Following the consent form, all participants would fill out the short survey for demographic information (gender, age and education). Then, similar for all participants, they completed the TMCT to induce cognitive depletion. Next, the natural context group would view natural pictures and conduct the digit span test, while the urban context group would view urban pictures and conduct the digit span test. The control group simply performed the digit span test without viewing pictures. Picture viewing took approximately five min, where each picture was shown for 7.5 s. For the two context groups, this study took approximately 35 min including setting up, signing informed consent and reading instructions. The control group was only asked to take the digit span test after making the TMCT, which took no longer than 25 min. To induce a somewhat similar setting for all participants, they were asked to place their laptop or monitor on a desk in front of them, in a quiet environment. In the context of investigating the effects of observing pictures of natural and urban environments on cognition, this research procedure shares close similarities to the procedures found in studies by Berman et al. [1], Gamble et al. [13] and Li and Sullivan [38]. The procedure of the present research is visually displayed in Figure 2.

### 2.4. Research Design

A between-subjects design was used with three experimental conditions: natural context, urban context and a control group. The natural context group contained 25 participants, the urban context group contained 24 participants and the control group contained 23 participants. The dependent variable was working memory capacity, which was defined by the mean number of correct forward and backward trials, as seen in Aumont et al. [48]. The separate forward and backward values are referred to as the two-error maximum length, the traditional measure of a participants forward or backward digit span. It is the highest amount of correctly recalled digits in one trial before making two consecutive mistakes on the following trials. 

### 2.5. Data Analyses

The digit span tests produced an individual data file per participant. Along with the information from the demographic survey and the corresponding experimental condition, each score of the digit span tests were manually entered in IBM SPSS per participant. After which, a mean score for the digit span performance was created. The dependent variable (mean digit span score) was separately inspected for normality and outlying cases using histograms and scatter plots. Cook’s distances were inspected, concerning influential cases would be removed (>1). Standardized residual scores were saved and inspected, concerning influential cases would be removed from the dataset (anything outside −3 and 3). 

To investigate the research question, a one-way ANOVA was performed to compare the effect of condition (nature, urban, and control) on working memory capacity (mean digit span performance). Afterwards, Bonferroni post hoc tests were conducted to investigate the hypothesis that working memory performance would be highest in the natural condition, followed by the control condition, and then the urban condition. The cut-off for significance was set at an alpha level of 0.05.

## 3. Results

Regarding outlier inspection, there were no scores which exceeded the Cook’s distance cutoff score of 1. There was one participant in the nature condition whose mean score of the digit span assessment exceeded the standardized residual value of 3 because of a high score. Deriving from the theory however, participants observing nature were expected to score higher on the digit span assessment, and since the maximum Cook’s distance of 1 was not violated, the participant was not excluded from further analyses. 

A one-way ANOVA of the digit span assessment scores (Figure 3) revealed a significant effect of condition, *F*(2, 69) = 5.59, *p* = 0.006, *ɳ_p_^2^* = 0.14. To find out which of the three conditions differed from each other, post hoc Bonferroni tests were conducted. The post hoc tests for multiple comparisons found that the mean digit span score was significantly different between the nature group (*M* = 6.86, *SD* = 1.22) and the urban group (*M* = 5.88, *SD* = 1.34), (*p* = 0.013, 95% C.I. = [0.17, 1.80]), and the nature group and the control group (*M* = 5.91, *SD* = 0.89), (*p* = 0.020, 95% C.I. = [0.12, 1.78]). In both cases, the nature group outperformed the urban and the control group significantly on the digit span assessment. At last, there was no statistically significant difference between the control group and the urban group, (*p* = 1.000, 95% C.I. = [−0.80, 0.87]). The average scores of the digit span assessment are visually displayed in Figure 3.

## 4. Discussion

The aim of this study was to investigate whether observing pictures of natural scenes compared to urban scenes would benefit working memory restoration after cognitive resources have been depleted. The hypothesis was that participants who looked at pictures of natural scenery after a working memory depleting task, would perform better at a subsequent working memory test (visual digit span test) than those who watched no pictures, followed by those who watched pictures of urban scenery. The present study closely connects to those found by Berman et al. [1], Gamble et al. [13] and Li and Sullivan [38] by showing significant positive effects of observing nature on the repletion of working memory capacity. In the present study however, the duration of the intervention was shorter than the intervention in the studies mentioned above. In addition, the current study also added a control condition in which nothing was viewed during the restoration phase.

Results from the current study indicated that participants who looked at natural environments for approximately five min were able to recall significantly more digits on the digit span assessment after making a cognitive depleting exercise, compared to participants who watched urban environments and participants in a control group who did not watch environments. This finding partly supported the proposed hypothesis that the natural context group would score higher than the urban context group and the control group. However, post hoc test results indicated that the control group did not significantly recall more digits than the urban group, which did not confirm the hypothesis. Post hoc results further indicated that participants in the natural context group recalled significantly more digits than those in the urban context group and the control group, which is in line with the results found in Berman et al. [1]. Moreover, the study by Gamble et al. [13] only found benefits regarding directed attention but not working memory when comparing nature and urban picture viewing. Though the current study did not investigate executive attention, it did find significant improvements in working memory capacity after viewing nature pictures, compared to urban pictures. Taken together, watching digitally presented pictures of natural environments does seem to favor the repletion of working memory capacity compared to watching urban environments or no environments after being depleted of cognitive resources. These findings are in line with the theoretical framework of the ART [7] and the SRT [23], where observing natural environments is expected to have restorative effects on cognitive resources, including working memory. Importantly though, both the ART and the SRT propose that observing urban environments can have the tendency to hinder repletion, or even further deplete cognitive resources.

In contrast to the studies by Berman et al. [1], Gamble et al. [13] and Li and Sullivan [38], the present study used a control group to investigate whether the previously found effect of watching nature scenery is caused by a positive effect of nature scenery, a negative effect of urban scenery, or both. Including a control group made this study more powerful, as to our knowledge, no previous research has used a control group to determine whether observing no pictures would be more beneficial on restoring the working memory compared to viewing urban pictures. Previous studies (e.g., [13]) could not differentiate whether the higher performance of the nature picture group was due to restorative effects of nature, or harmful effects or urban pictures. In the present study, results clearly indicate no significant differences between the urban context group and the control group in neither the digit span forward or backward trials. It can therefore be stated that urban stimuli were neither significantly harmful nor beneficial to cognitive restoration. Additionally, this study found improvements after viewing pictures for only approximately five min, whereas in the Berman study [1], picture viewing took 10 min. The latter finding indicates that even a short intervention of watching natural scenery for only 5 min can benefit the restoration of the working memory.

It is important to note that though significant results in favor of watching nature pictures have been found, these results should be interpreted with caution. Since all participants had the chance to participate from home, the experiment did not take place in a well-controlled setting. Even though all participants received identical instructions to make the tests in a quiet environment, on a laptop or monitor placed in front of them, it is not possible to confirm this was always the case. On the other hand, using this setup it could be argued that even in varying contexts, taking a break to view natural environments can be beneficial for the working memory capacity after a cognitive demanding activity. Therefore, the benefits found in the nature condition may not be due to these varying contexts in which each participant engaged with experiment. Additionally, the depleting task and the digit span were both presented visually to the participants. It is therefore unclear whether the performance on the digit span was influenced because it was presented visually and not auditory. In practice, the findings still hold relevance since many cognitive activities require visual information to be processed.

The current research is not without its own limitations. The participants used in the present study were university students aged between 18 and 30 years old. Whether the restorative effects of natural stimuli are generalizable to other groups is not clear. Though previous research by Li and Sullivan [38] found a positive impact of observing nature on working memory in high school students. Whereas a study by Amicone et al. [49] found positive effects of primary school children interacting with physical nature on working memory. Additionally, the present study took place in the city of Rotterdam, which is a considerable urban environment. It can be speculated that participants living in such urban environments are more used to urban environmental arousal and therefore feel more comfortable with it. Due to this, viewing urban pictures may not have been depleting in the present study. This speculation derives from a study by Lederbogen et al. [50] which found that people who were exposed to different types of environments over the long term react differently to stressful tasks, which in turn could influence cognitive performance. Another limitation of this study is that the working memory capacity was tested immediately after observing either natural or urban pictures. It is unclear how long the benefits of observing nature actually lasts, and whether observing nature as an intervention in academic setting would result in improved academic performance on the long term. At last, this study primarily tried to use the theoretical framework to investigate whether the restorative effect of natural stimuli on cognition actually occurs. The casual mechanisms that are responsible for the observed effects were not investigated, neither were more specific types of executive functions which could have benefitted from observing nature. 

A possible confound in the study might be that the pictures of the natural scenes have a different composition of colors than those of the urban scenes. Where the natural scenes are dominated by green shades, the urban scene pictures are more dominated by grey and brown shades. Previous research seemed to indicate that color can have an effect on cognitive task performance. For example, Mehta and Zhu [51] found that a red colored screen background improved task performance in which details were important, while a blue background improved creative task performance. This result was replicated by Lichtenfeld, et al. [52], who also showed a positive effect of green compared with white, grey, red and blue. Important to mention is that in our study, the pictures were used as isolated stimuli separately from the cognitive tasks, while in the studies described above, the colors co-occurred with the task stimuli (background colors). We suggest this could have influenced the task performance differently, as we focused on WM restoration after being fatigued and the other studies more on the influence of colors during task performance. Also, the studies above used different tasks than ours and did not focus on working memory depletion. However, it would be interesting for future research to investigate whether the natural scenery specifically has restorative effects, or whether the color green can have a similar effect.

Some previous studies have investigated the restorative effects of nature on cognition by having participants physically present in nature, mainly by taking a walk. For instance, Bratman and colleagues [6] had participants walk in either a nature or urban setting for 50 min, after which cognitive tests were taken, and found significant positive results on cognition after walking in nature. It is hard to implicate such findings in actual classrooms, as parks may not be accessible for all schools and taking a walk would take up precious classroom time. Though showing videos with sound could imitate actual nature, the present study shows how even looking at pictures brings benefits. Therefore, results from the current study suggest that the short intervention we used could be implemented in an academic setting. When taking in-class breaks for example, teachers could set the screensaver of a smartboard to a slideshow of nature pictures, where students could then be passively exposed to an artificial natural environment. Additionally, textbooks or online learning materials could fill blank spaces with pictures of natural environments, again passively exposing students to natural environments.

Passive exposure to natural environments could be enough evoke restorative cognitive effects when engaging in classroom activities according to a systematic review by Norwood et al. [8]. Moreover, Moreno and colleagues [30] created an app which exposed students to videos of natural environments for two minutes. Though higher app usage led to higher academic performance, executive functions, such as working memory, were not associated. The idea of using such an app in the classroom to improve or restore cognitive processes is favorable. However, two minutes of exposure might not have been enough to evoke the desired effects. Using a slideshow variant, instead of a video, for a slightly longer period of time could be beneficial, as discovered in the present study. These implications are only applicable to academic environments which have access to devices to display artificial natural environments. Having plants or wall pictures present in classrooms may not be targeted enough for students to be exposed. Nonetheless, adding nature environmental stimuli to a classroom is a practical, achievable and low-budget option to possible aid students in restoring cognitive capacities after engaging in demanding classroom activities. Even though long-term effects are yet to be investigated, the immediate effects of simply looking at nature are present in the current study. Since the intervention of looking at natural environments would take no more than five minutes, this could practically be integrated in regular classroom activities.

The implication of the findings from the present study may also be relevant beyond academic settings, such as the workplace. People who have cognitively demanding jobs (e.g., effortful computer jobs involving complex systems, information technology, financials or jobs that require constant monitoring) may eventually experience some form of cognitive depletion. A short intervention of observing natural scenery which can help restore cognitive resources in the workplace may therefore be highly appreciated, and result in increased productivity. In addition to previously made points regarding nature not being easily accessible for everyone, observing digitally presented nature may also be relevant in non-professional or private settings for people seeking a break for any activity, which may deplete cognitive resources, such as learning in general, practicing music, and multimedia editing.

Future research on the topic should focus on more types of executive functions or cognitive mechanisms which might be restored by exposure to nature. This study primarily focused on working memory, which in turn could be associated with higher academic performance [28,29]. When future research would find more cognitive components which may benefit from observing nature, the implication of nature in academic setting would become more powerful and desirable. Especially in an academic setting, inhibitory control would be a desirable executive function to investigate in a similar context. Inhibitory control concerns the control over automatic urges and shifts of attention by distractions. If a short nature viewing intervention would benefit inhibitory control, people would be able to stay focused on the subjects at-hand for a longer period of time. To our knowledge, no extensive research has been conducted regarding the benefits of exposure to nature on inhibitory control, though a neuroimaging study [53] does suggest preliminary evidence of improved inhibitory control after exposure to nature. Additionally, selective attention and directed attention both seem to show potential in terms of improvement after a short exposure to natural environments, as shown in a review by Vella-Brodrick and Gilowska [24]. With multiple executive functions proving to benefit from nature exposure, future research could focus on how these benefits impact actual learning and performance in the classroom. Therefore, research which integrates natural scenes in actual classrooms should be conducted to investigate whether the restorative effects of nature still hold value in practical setting. Beside the cognitive components, future research could further investigate specific characteristics of natural environments which may bring the desired effects. For example, whether the colors represented in natural environments (e.g., green) might be beneficial, or even the presence of plants in any scene. In light of the limitation regarding long term academic impact of observing nature, future research could implement a brief nature-observing intervention in academic setting, to see whether observing nature for short periods of time during class has any impact on academic performance at all. 

## 5. Conclusions

In conclusion, we found restorative effects of working memory after observing digitally presented natural scenery for five min. Measured using both the digit span forward and backward trials taken together, participants were able to recall more digits after observing natural scenes while being depleted of cognitive resources. These results were in line with and extended those of previous studies by Berman et al. [1] and Gamble et al. [13] on the restorative effects of nature on cognition. Urban scenery proved to be neither helpful nor harmful in the present sample. These findings provide a foundation for an easy and low-cost intervention to benefit the restoration of the working memory capacity on the short term. These types of interventions could be integrated in academic setting in order to facilitate students in classroom performance. 

## Figures and Tables

**Figure 1 ijerph-20-00188-f001:**
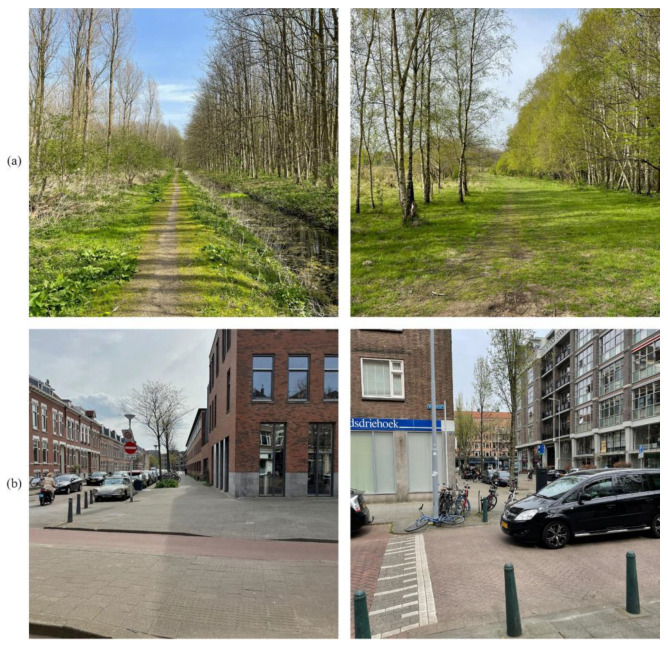
The first row of pictures (**a**) indicates a sample of two pictures used in the nature condition, the second row of pictures (**b**) indicates a sample of two pictures used in the urban condition.

**Figure 2 ijerph-20-00188-f002:**
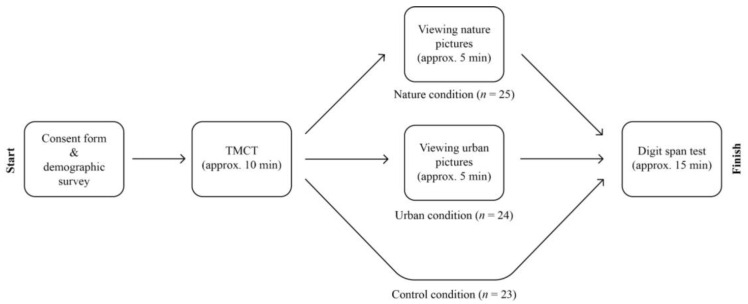
Flowchart of the Research Procedure. TMCT = Trier Mental Challenge Test.

**Figure 3 ijerph-20-00188-f003:**
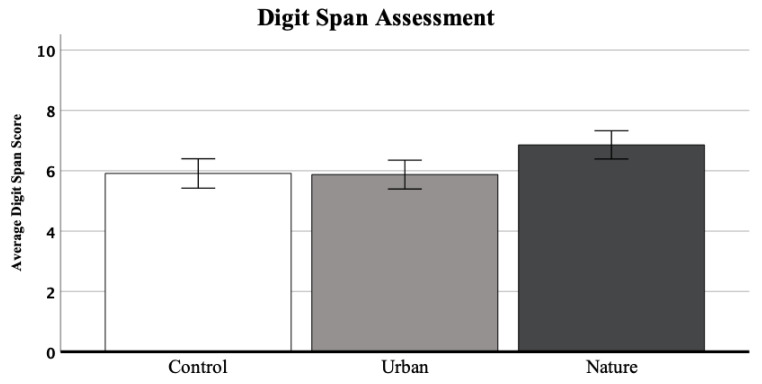
Mean Digit Span Score per Restorative Condition; Control, Urban, and Nature.

## Data Availability

The raw data supporting the conclusions of this article will be made available by the authors, without undue reservation.

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
