# Peer review of "Restorative Effects of Observing Natural and Urban Scenery after Working Memory Depletion"

_ijerph, 2022, doi:10.3390/ijerph20010188_

Round 1

Reviewer 1 Report

This is an adequate paper for a typical study of this kind.  There are no issues with study design, background, methods, results or conclusions.  At the same time, there is nothing novel about this study that advances the field.  Authors may want to spend more time considering how this manuscript could be updated to advance the research rather than merely replicating it. 

Author Response

Response to Reviewer #1

Manuscript ‘Restorative Effects of Observing Natural and Urban Scenery after Working Memory Depletion’

Reviewer #1
This is an adequate paper for a typical study of this kind.  There are no issues with study design, background, methods, results or conclusions

C1: There is nothing novel about this study that advances the field.  Authors may want to spend more time considering how this manuscript could be updated to advance the research rather than merely replicating it. 

R1: Thank you for this comment. However, we would like to emphasize the different novel aspects of our study that we think may advance the field. Firstly, previous research on the effects of natural vs. urban scenery has been based on the attention restoration theory using attention tasks as the main measure of effect. In the present study we focused specifically on working memory depletion, which can be considered as new insight in this research context. Secondly, research that did use working memory as a measure of effect in the context of cognitive resource depletion, did never consider natural vs. urban scenery as influencers of resource restoration (e.g., Chen et al., 2017; Healey et al., 2011; Schmeichel, 2007). Thirdly, unlike previous studies on the attention restoration effect, we used a control group without an intervention. In previous studies, researchers only compared restoration effects of natural scenery with urban scenery, which makes unequivocal interpretation of the results impossible (e.g., Berman et al., 2008; Gamble et al., 2014; Li & Sullivan, 2016). By using a control group, we were able to differentiate whether the observed effects were due to the restorative effects of observing nature, or the possible restorative hindering effects of observing urban scenery, as proposed by the attention restoration theory and the stress reduction theory.

Further clarification has been added under heading 1.5. (page 4).

Berman, M. G., Jonides, J., & Kaplan, S. (2008). The cognitive benefits of interacting with nature. Psychological Science, 19(12), 1207–1212. https://doi.org/10.1111/j.1467-9280.2008.02225.x

Chen, O., Castro-Alonso, J. C., Paas, F., & Sweller, J. (2017). Extending cognitive load theory to incorporate working memory resource depletion: Evidence from the spacing effect. Educational Psychology Review, 30(2), 483–501. https://doi.org/10.1007/s10648-017-9426-2

Gamble, K. R., Howard, J. H., & Howard, D. V. (2014). Not just scenery: Viewing nature pictures improves executive attention in older adults. Experimental Aging Research, 40(5), 513–530. https://doi.org/10.1080/0361073x.2014.956618

Healey, M. K., Hasher, L., & Danilova, E. (2011). The stability of working memory: Do previous tasks influence complex span? Journal of Experimental Psychology: General, 140(4), 573–585. https://doi.org/10.1037/a0024587

Li, D., & Sullivan, W. C. (2016). Impact of views to school landscapes on recovery from stress and mental fatigue. Landscape and Urban Planning, 148, 149–158. https://doi.org/10.1016/j.landurbplan.2015.12.015

Schmeichel, B. J. (2007). Attention control, memory updating, and emotion regulation temporarily reduce the capacity for executive control. Journal of Experimental Psychology: General, 136(2), 241–255. https://doi.org/10.1037/0096-3445.136.2.241

Reviewer 2 Report

I found this paper to be a joy to read and a really interesting addition to the growing body of work on the benefits of simulated nature.  The choice of the digit span task as a measure of working memory capacity is entirely appropriate and the use of a reliable and valid measure of cognitive depletion was well considered.  I had a question about how consistency in the level of cognitive depletion was assessed across the three groups as this did not emerge clearly for me from the paper and I would like this to be clarified in the method or results.  This may be because the authors identify that no data were collected from the Trier Mental Challenge Test, and this would suggest they had not been able to establish a baseline level of depletion. Therefore, this should be clarified.  I think it is likely that the authors have considered this (and indeed that the TMCT is designed to do this in a standardised way) and that the clarification will be a matter of a few sentences.  However, if this was not fully established a discussion of the potential implications of this would be useful to add to the discussion. 

The other thing which would be useful to extend in the discussion is the consideration of future research around tests of executive function.  I felt that this section could benefit from some further additions.  However aside from this, I did not see the necessity for a significant amount of work. 

Stylistically the authors need to review some of their larger paragraphs for clarity - splitting some of these into smaller paragraphs would enhance readability - particularly for an international audience.  However other than this I felt the paper was exceptionally well written (I could not even spot typos!) and the authors are to be commended on this attention to detail.  

Overall a well considered and executed study which has potential real world impact in terms of the academic interventions identified by the authors.  It could have further benefits in workplaces where detail-oriented and cognitive depleting job roles are present (e.g. anything involving review of computer code, financial systems or video/CCTV footage) and it would be interesting for the authors to consider future research outside of the academic environment and in this type of setting. 

Author Response

Response to Reviewer #2

Manuscript ‘Restorative Effects of Observing Natural and Urban Scenery after Working Memory Depletion’

Reviewer #2

C1: I had a question about how consistency in the level of cognitive depletion was assessed across the three groups as this did not emerge clearly for me from the paper and I would like this to be clarified in the method or results.  This may be because the authors identify that no data were collected from the Trier Mental Challenge Test, and this would suggest they had not been able to establish a baseline level of depletion. Therefore, this should be clarified.  I think it is likely that the authors have considered this (and indeed that the TMCT is designed to do this in a standardised way) and that the clarification will be a matter of a few sentences.  However, if this was not fully established a discussion of the potential implications of this would be useful to add to the discussion. 

R1: Firstly, we thank you for the insightful comments and suggestions. We agree with the comment regarding further clarification of the stressor task. Clarification has been provided as well as another literature source which describes the TSST and variations as a highly standardized test to induce physiological and psychological stress (resulting in cognitive fatigue) in a consistent and reliable way (page 5).

C2: Another thing which would be useful to extend in the discussion is the consideration of future research around tests of executive function.  I felt that this section could benefit from some further additions.

R2: We have now extended this part in the discussion. Another suggestion for future research regarding specific executive functions have been added. As well as an overarching suggestion to investigate how current findings on improved executive functions after nature exposure translate themselves to actual improvements in learning and performance (page 12).

C3: Stylistically the authors need to review some of their larger paragraphs for clarity - splitting some of these into smaller paragraphs would enhance readability - particularly for an international audience.

R3: Thank you for pointing this out to us. Various (longer) paragraphs have been split up at appropriate points to increase readability.

C4: It could have further benefits in workplaces where detail-oriented and cognitive depleting job roles are present (e.g. anything involving review of computer code, financial systems or video/CCTV footage) and it would be interesting for the authors to consider future research outside of the academic environment and in this type of setting. 

R4: Agreed. Further implications have been added extending the implication to the workplace, and also non-professional settings which may be cognitively demanding (page 12).